# MafB, WDR77, and ß-catenin interact with each other and have similar genome association profiles

Lizhi He[1☯], Mingshi Gao[2☯], Henry Pratt[2☯], Zhiping Weng[2]*, Kevin Struhl[1]*

**1** Dept. Biological Chemistry and Molecular Pharmacology, Harvard Medical School, Boston, MA, United states of America, **2** Program in Bioinformatics and Integrative Biology, University of Massachusetts Medical School, Worcester, MA, United states of America

☯ These authors contributed equally to this work.
* kevin@hms.harvard.edu (ZW); zhiping.weng@umassmed.edu (KS)

**Data Availability Statement:** All sequencing data were deposited on National Cancer for Biotechnology Information Gene Expression Omnibus (GEO). GSE195740 is the accession number for all the data, with GSE1957437 being

## Abstract

MafB (a bZIP transcription factor), ß-catenin (the ultimate target of the Wnt signal transduction pathway that acts as a transcriptional co-activator of LEF/TCF proteins), and WDR77 (a transcriptional co-activator of multiple hormone receptors) are important for breast cellular transformation. Unexpectedly, these proteins interact directly with each other, and they have similar genomic binding profiles. Furthermore, while some of these common target sites coincide with those bound by LEF/TCF, the majority are located just downstream of transcription initiation sites at a position near paused RNA polymerase (Pol II) and the +1 nucleosome. Occupancy levels of these factors at these promoter-proximal sites are strongly correlated with the level of paused Pol II and transcriptional activity.

## Introduction

Transient activation of the Src oncoprotein in a non-transformed breast cell line (MCF-10A) mediates an epigenetic switch to a stably transformed state [1]. This epigenetic switch is mediated by a positive feedback loop whose maintenance results in a chronic inflammatory state required for transformation. This loop involves inflammatory transcription factors NF-κB, STAT3, and AP-1 proteins that form complexes that directly regulate hundreds of genes in oncogenic pathways [1–4]. This regulatory network functions coherently in many cancer types, and it provides the basis for a cancer inflammation index that classifies cancer types in a manner distinct from mutation or developmental origin [4].

More than 40 DNA-binding transcription factors contribute to the transformed state in this inducible model of breast cellular transformation, with potential target sites identified via DNA sequence motifs [5]. In addition, cellular transformation depends on transcriptional co-activators that are key components of signal transduction pathways and are recruited to target sites by DNA-binding proteins. Such co-activators include the YAP and TAZ paralogues, the ultimate targets of the Hippo signaling pathway [6], and the calcium-binding cytokines S100A8 and S100A9 [7].

the subset for the ChIP-seq data and GSE195739 for the RNA-seq data.

**Funding:** HHS | National Institutes of Health (NIH): Lizhi He, Kevin Struhl CA107486; HHS | National Institutes of Health (NIH):Mingshi Gao,Henry Pratt, Zhiping Weng HG009446

**Competing interests:** No, there is no conflict of interest. My manuscript contains the following statement: "The authors declare that they have no conflict of interest."

Here, we describe an unexpected connection between MafB, WDR77, and ß-catenin. MafB is a cousin of AP-1 transcription factors, with related but not identical DNA-binding specificity [8]. It ranked 6[th] on an ordered list of candidate transcription factors involved in transformation, and indeed MafB depletion reduces transformation in this model [5]. In addition, a few reports link MafB to cancer [9–11].

WDR77 is a transcriptional co-activator of the androgen and other hormone receptors [12]. Overexpression of WDR77 stimulates tumorigenesis in breast and other cancer types [13,14]. WDR77 is also a subunit of the 20S methylosome complex that includes PRMT5, an enzyme that symmetrically di-methylates arginines in spliceosomal Sm proteins and histones [15].

ß-catenin, the ultimate effector of the Wnt signaling pathway, plays critical roles in development and diseases including cancer [16–19]. ß-catenin is a major component of adherens junctions at the cell membrane where it regulates cell growth and adhesion. Upon activation of the Wnt pathway, ß-catenin translocates to the nucleus and acts as a transcriptional co-activator, primarily via its interaction with the LEF/TCF family of DNA-binding transcription factors.

Here, we show that MafB, WDR77, and ß-catenin are important for transformation, interact directly with each other, and have similar genomic binding profiles. Surprisingly, the majority of common MafB/WDR77/ß-catenin target sites are located just downstream of transcription initiation sites at a position near paused RNA polymerase (Pol) II and the +1 nucleosome. Occupancy levels of these factors at these promoter-proximal sites are strongly correlated with the level of Pol II and transcriptional activity.

## Results

### The initial connection between MafB, WDR77, and ß-catenin

Our inducible model of breast cellular transformation is based on a derivative of the non-transformed cell line MCF-10A that expresses ER-Src, a fusion between the v-Src oncoprotein and the ligand-binding domain of estrogen receptor [20,21]. Treatment of these cells with tamoxifen activates v-Src and triggers an epigenetic switch between non-transformed and transformed cells. This work began with our interest in transcriptional co-activators involved in cellular transformation in this model. We previously analyzed the YAP and TAZ paralogs that are the ultimate effectors of the Hippo signaling pathway [6], as well as the calcium-binding cytokines S100A8 and S100A9 [7].

Here, we initiated studies on ß-catenin, the ultimate target of the Wnt signaling pathway that acts as a transcriptional co-activator of the LEF/TCF transcription factors and has many connections to cancer [16–19]. Using mass spectrometry, we identify many proteins interacting with ß-catenin, one of which is WDR77. In seemingly unrelated work, we began to further analyze MafB, a cousin of AP-1 transcription factors, that we identified as a strong candidate for being involved in transformation [5]. As discussed below, ChIP-seq experiments revealed a striking similarity in genomic binding profiles of MafB, WDR77, and ß-catenin.

### MafB, WDR77, and ß-catenin are important for transformation

Using CRISPR-Cas9, we generated derivatives of the ER-Src cell line individually knocked out for MafB, WDR77, or ß-catenin (Fig 1A). We examined their effect on transformation by their ability to grow in conditions of low attachment in the presence of tamoxifen [22]. All three knockout cell lines show a substantial defect in cell growth in low attachment conditions as compared to the parental cell line (Fig 1B). In contrast, the knockout and parental cell lines show similar growth under conditions of high attachment (Fig 1C). In addition, siRNA

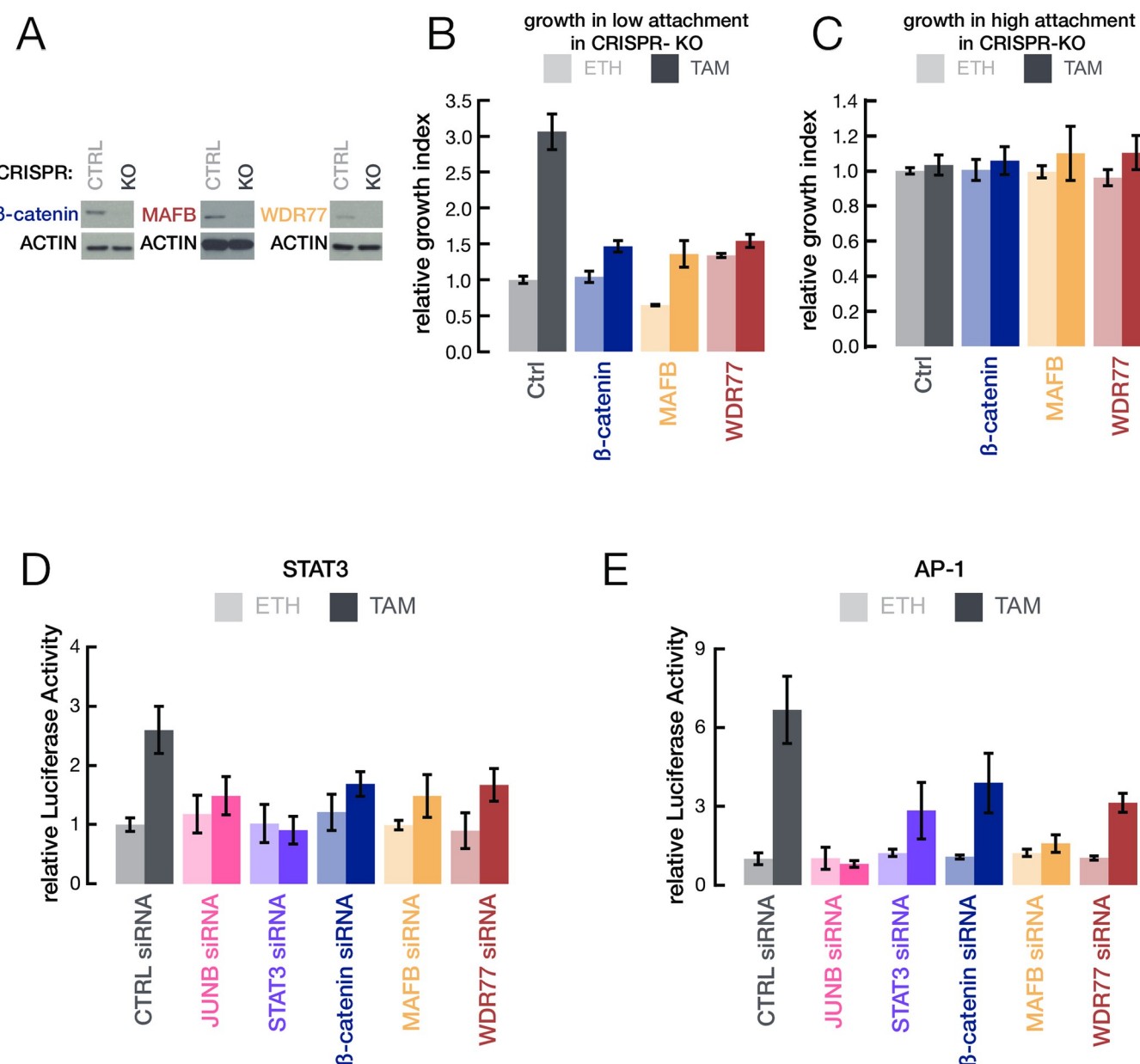

**Fig 1. β-catenin, MafB, and WDR77 are important for transformation.** (A) Western blots of β-catenin, MafB, and WDR77 in CRISPR knocked-out (CRISPR-KO) and CRISPR control cells (CRISPR-Ctrl). (B, C) Relative growth in low attachment (B) or normal (high-attachment)(C) conditions of transformed (tamoxifen treated; TAM shown in darker colors) and non-transformed (ethanol treated; ETH shown in lighter colors) cells with the indicated knockouts. (D, E) Relative STAT3 (D) or AP-1 (E) reporter (luciferase) activity in non-transformed and transformed cells upon the indicated siRNA knockdowns. Error bars indicated ± SD of 3 replicates.

depletion experiments indicate that all three of these proteins reduce transcriptional induction of STAT3 (Fig 1D) and AP-1 (Fig 1E) reporter constructs under transformation conditions. Thus, MafB, WDR77, and ß-catenin are important for transformation.

## MafB, WDR77, and ß-catenin physically interact with each other

As mentioned above, WDR77 was identified as a ß-catenin-interacting protein by immuno-precipitation followed by mass spectrometry. To confirm this and the other interactions, we

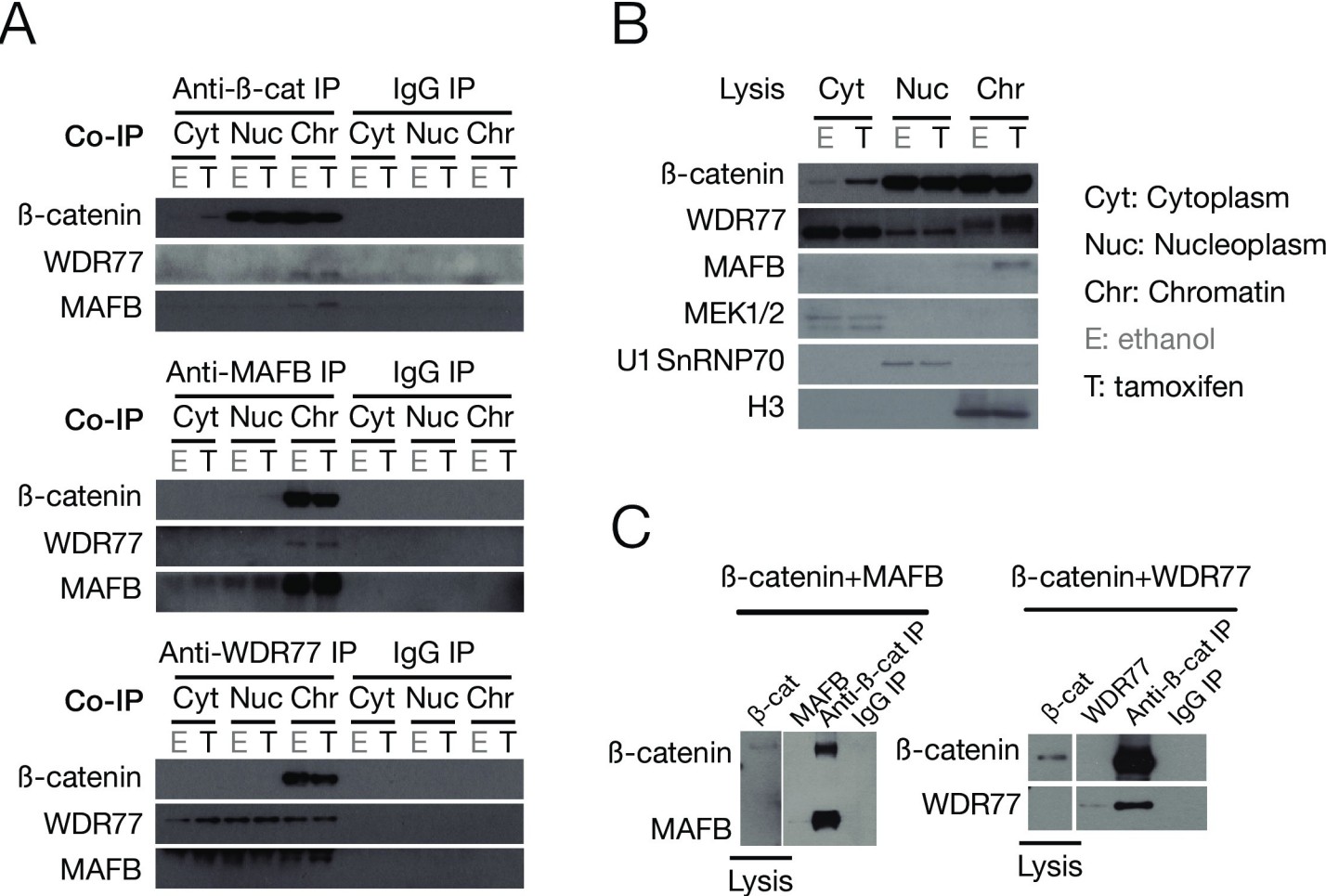

**Fig 2. The direct interaction of β-catenin, MafB, and WDR77.** (A) Co-IP of endogenous β-catenin, MafB, and WDR77 proteins in the cytoplasm (Cyt), nucleoplasm (Nuc), and chromatin (Chr) fractions of transformed (T) and non-transformed (E) cells. IgG was used as a co-IP control. (B) Western blots of the indicated proteins in fractions from the cell-free extract prior to the co-IP. Fractions are defined by MEK1/2 (cytoplasm), U1 SnRNP70 (nucleoplasm), and H3 (chromatin). (C) co-IP of recombinant β-catenin, MafB, and WDR77 proteins.

performed co-immunoprecipitation (co-IP) in cytoplasm, nucleoplasm, and chromatin fractions in non-transformed and transformed cells. Each individual protein co-immunoprecipitated the other two proteins in the chromatin fraction (Fig 2A and 2B). This indicates that these three proteins interact with each other, although not necessarily in the same complex. The interactions of ß-catenin with WDR77 and MafB are direct because they can be observed by coimmunoprecipitation of histidine-tagged recombinant proteins generated in *E. coli* (Fig 2C).

## MafB, WDR77, and ß-catenin associate with common genomic loci

We performed ChIP-seq to identify the genomic target sites of MafB, WDR77, and ß-catenin in both non-transformed and transformed cells. Binding by ß-catenin (and to a lesser extent MafB) is increased in transformed cells, whereas binding by WDR77 occurs at similar levels in non-transformed and transformed cells (Fig 3A). We do not understand why transformed cells show increased ß-catenin binding to target sites but not total chromatin binding; perhaps this reflects post-translational modification of ß-catenin in transformed cells. Strikingly, and

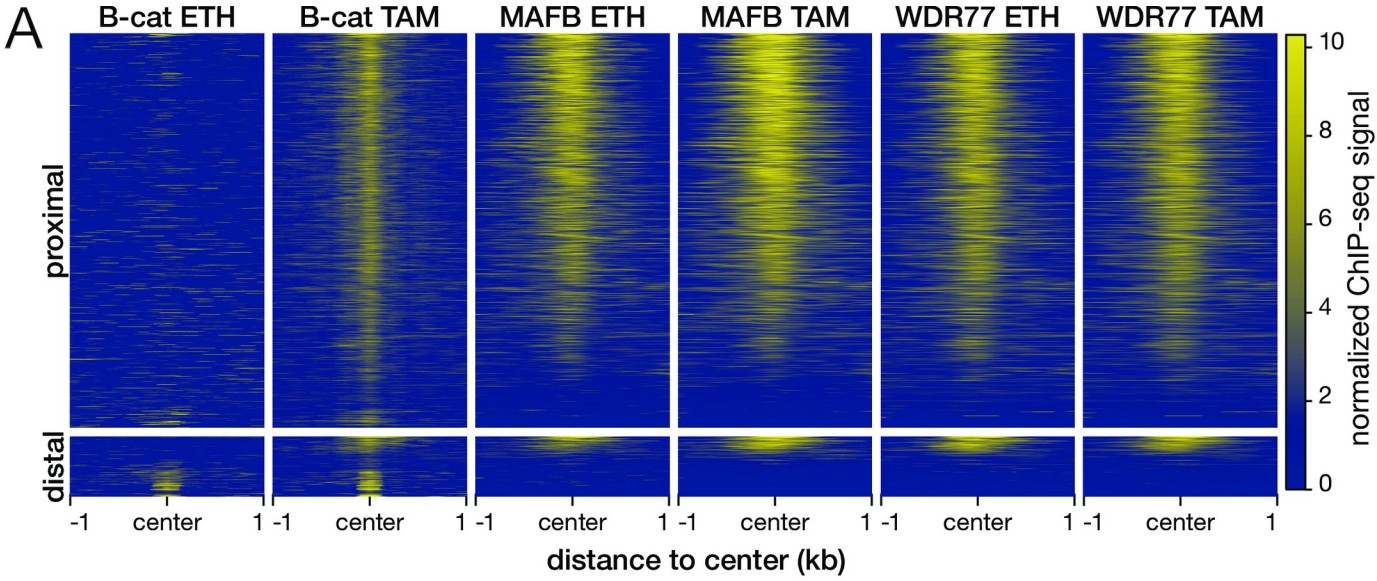

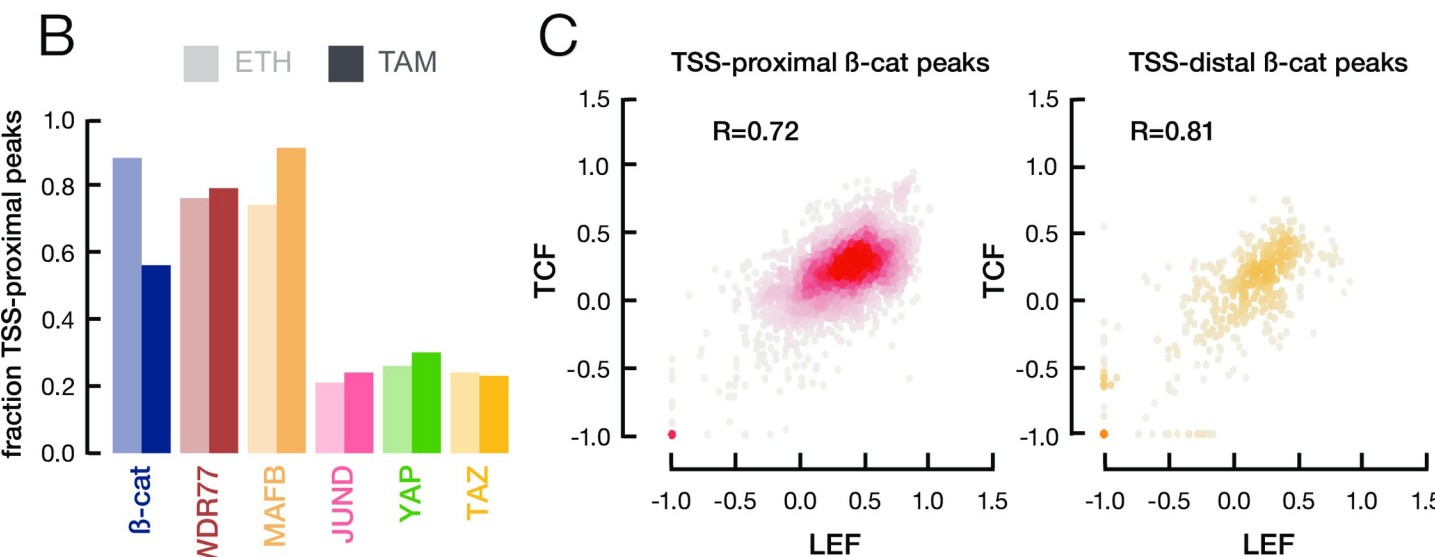

**Fig 3. Similar genome association profiles of ß-catenin, MafB, and WDR77.** (A) Heatmap of ChIP-seq signals of the indicated proteins in tamoxifen- (TAM) or ethanol- (ETH)-treated cells. Each row of the heatmap corresponds to a genomic region bound by ß-catenin in transformed cells, with rows sorted by the average ChIP-seq signal across six experiments. TSS-proximal and TSS-distal peaks are shown separately. (B) Percentage of ChIP-seq peaks of the indicated proteins that are TSS proximal. (C) Correlation of TCF and LEF ChIP-seq signals at TSS-proximal and TSS-distal peaks of ß-catenin in transformed cells-TAM peaks. Pearson correlation coefficients are shown.

in accord with their physical interactions, these three proteins bind to many common genomic locations (Fig 3A). As will be discussed in more detail below, most of these common target loci are located near the transcriptional start site (TSS), unlike the preferred distal locations for the AP-1 transcription factor JUND and the YAP and TAZ co-activators that are important for transformation (Fig 3B). As expected from its interaction with LEF/TCF family of transcription factors, ß-catenin target sites are bound by LEF and TCF to comparable extents (Fig 3C).

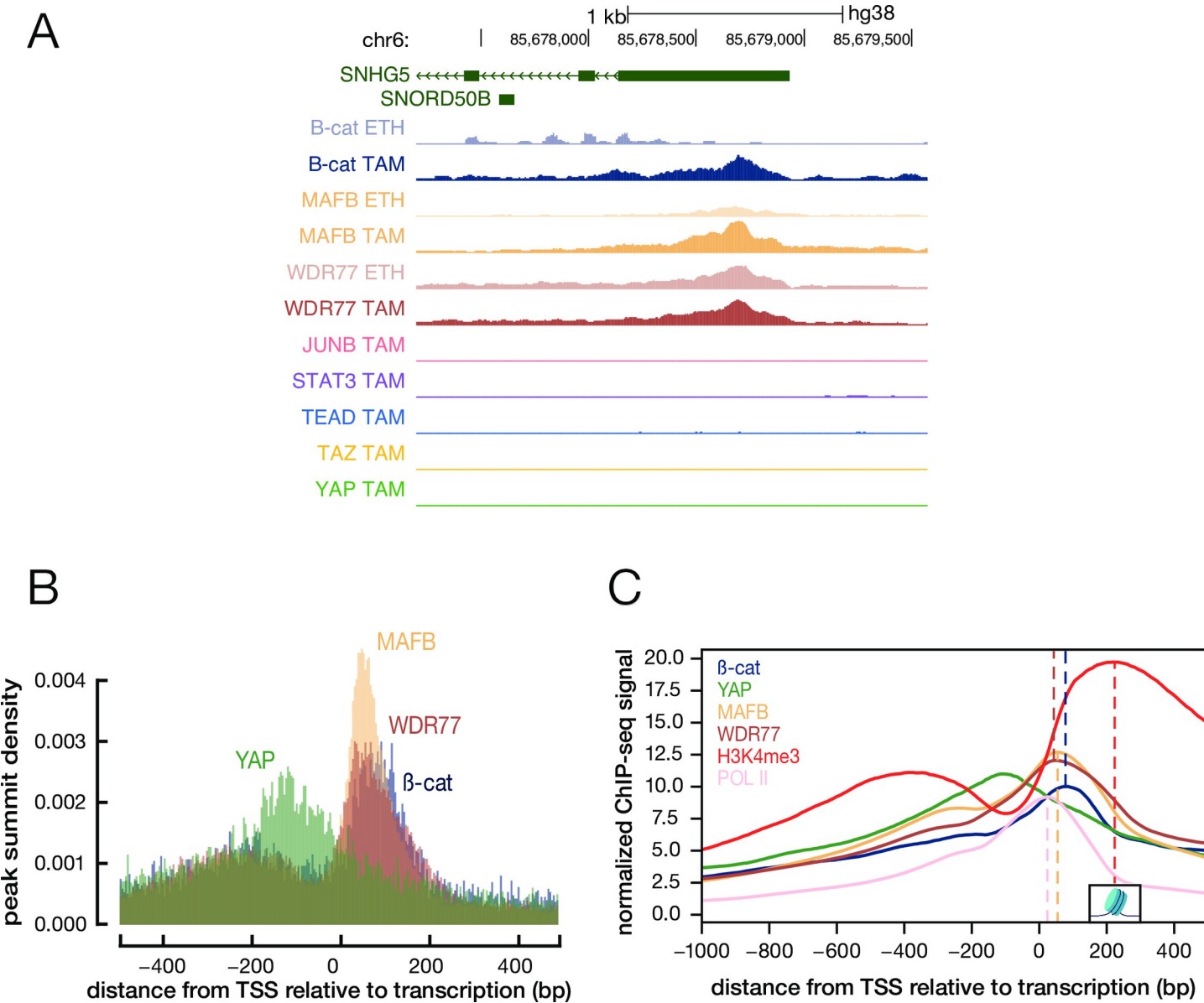

**Fig 4. Many ß-catenin/MafB/WDR77 sites are located between paused Pol II and +1 nucleosome.** (A) Genome browser view of ChIP-seq signals of the indicated proteins in non-transformed (ETH) or transformed (TAM) calls at an example locus (*SNHG5*). (B) Histogram of ChIP-seq peak summits around the TSS in transformed cells. (C) Aggregated ChIP-seq signal with the regions between -1000 and +500 with respect to the TSS. Dashed line shows the summit (position with the highest aggregated signal), and the +1 nucleosome is centered at the summit of the H3K4me3 signal.

### Many MafB/WDR77/ß-catenin target sites are located just downstream of paused Pol II

Unexpectedly, many of the TSS-proximal ß-catenin, MafB, and WDR77 target sites are located downstream of the TSS (Fig 4, with one example locus shown in panel A). This location is in marked contrast to the location of target sites for YAP (Fig 4B) and numerous transcription factors that are typically located 50–200 bp upstream of the TSS. Peak summit analysis indicates that the ß-catenin, MafB, and WDR77 target sites are located between paused Pol II and the +1 nucleosome, which is defined by associated chromatin mark H3-K4me3 (Fig 4B and

4C). MafB and WDR77 peaks coincide about 25 bp downstream of paused Pol II, whereas ß-catenin peaks are located ~40 bp further downstream (Fig 4C).

These unexpected results made us consider the possibility of an experimental artifact. In this regard, the ß-catenin, MafB, and WDR77 ChIP-seq experiments involved a two-step procedure in which protein-protein crosslinking with a mixture of homobifunctional reagents with different spacer lengths was followed by the formaldehyde treatment to link proteins to DNA. However, multiple lines of evidence argue against an artifact. First, ß-catenin association was observed in chromatin from transformed cells but not non-transformed cells (Figs 3A and 4A). Second, the antibodies for ß-catenin (rabbit), MafB (goat), and WDR77 (mouse) come from different species. Third, biological replicates that involved different chromatin samples gave reproducible results. Fourth, using the same procedure and sometimes the same chromatin samples, the unexpected pattern was not observed in ChIP-seq experiments for YAP, TAZ, or TEAD [6] or other proteins (e.g. the example locus in Fig 4A). Fifth, the pattern observed for ß-catenin, MafB, and WDR77 is quite specific, and it is clearly different from that of a previously described artifact in which virtually any nuclear protein associates weakly with highly transcribed coding regions [23,24].

## MafB/WDR77/ß-catenin association near the Pol II pause site is correlated with, but does not affect, transcriptional activity

To identify genes regulated by MafB, WDR77, or ß-catenin, we individually depleted these factors by siRNA-mediated knockdown in tamoxifen-treated cells. RNA-seq analysis identifies ~3000 differentially expressed genes for each depletion as compared with a control siRNA. Roughly equal number of genes show increased or decreased expression, and the transcriptional profiles show significant overlap between differentially expressed genes (Fig 5A and 5B). These transcriptional profiles also resemble those observed upon depletion of various DNA-binding proteins that affect transformation efficiency, reflecting the positive feedback loop required for transformation [5].

Genes with MafB/WDR77/ß-catenin binding near paused Pol II have dramatically higher levels of expression than observed in the overall population of genes (Fig 5C). In addition, the level of MafB/WDR77/ß-catenin binding at such sites is correlated with the level of paused Pol II (R = 0.56 for MafB, 0.61 for WDR77, and 0.21 for ß-catenin). However, there is no significant correlation between MafB/WDR77/ß-catenin binding near paused Pol II and differential transcription in cells depleted of any of these factors (Fig 5D).

## Discussion

ß-catenin [16–19], MafB [9–11], and WDR77 [13,14] have all been linked to cancer, and they are important for breast cellular transformation in our Src-inducible model. These three proteins, previously unrelated by any functional criterion, interact directly with each other, and they have similar genomic binding profiles. Unexpectedly, most common target sites map downstream of the TSS between paused Pol II and the +1 nucleosome, a location not previously associated with any other protein.

How are ß-catenin, MafB, and WDR77 recruited to the region between paused Pol II and the +1 nucleosome? It seems unlikely that they are recruited by Pol II or the +1 nucleosome (with its modifications such as H3-K4me3) because their peak summits do not co-localize, unlike other situations in which co-activators and co-repressors are recruited by DNA-binding proteins [25]. In addition, these three proteins are not associated with other genomic regions bound by Pol II or nucleosomes, and the ChIP-seq signals are not observed by formaldehyde crosslinking alone. However, occupancy levels of the three proteins are strongly correlated

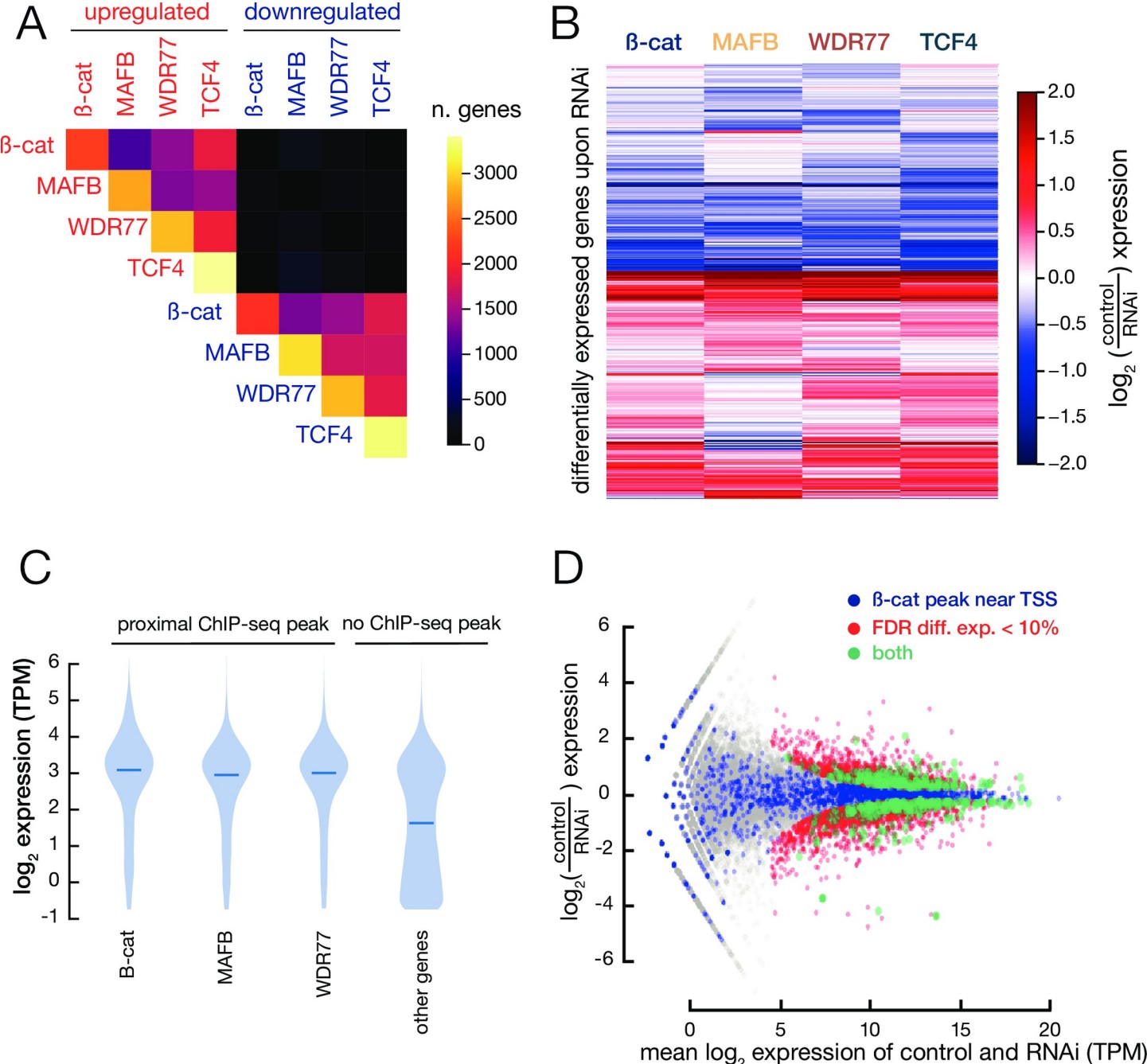

**Fig 5. ß-catenin/MafB/WDR77 binding is correlated with transcriptional activity.** (A) Number of genes shared between upregulated (top and left) and downregulated (bottom and right) gene sets following RNAi against the indicated proteins. The top-right black square (near-zero shared genes) indicates that the directionality of expression change is highly conserved. (B) Log₂ expression fold-change of individual protein-coding genes (rows of the heatmap) between control and RNAi knockdown. (C) Violin plot showing expression levels in transformed cells of sets of genes defined as with or without proximal ß-catenin/MafB/WDR77 ChIP-seq peaks. (D) MA plot showing the gene expression levels of ß-catenin RNAi and control. Blue dots indicate genes whose TSSs are bound by ß-catenin, red dots indicate differentially expressed genes (FDR < 10%), and green dots indicate differentially expressed genes that are bound by ß-catenin (both). The few purple dots indicate that ß-catenin binding is not causally associated with significant changes in gene expression after RNAi knockdown.

with the level of Pol II association at the pause site. This observation suggests that recruitment of ß-catenin, MafB, and WDR77 involves some component related to Pol II pause release. This component could be a protein(s) that is part of the pause release mechanism or a DNA or

RNA structure that occurs at this position upon pause release. A more speculative possibility is that the region between paused Pol II and the +1 nucleosome is involved in the formation of a biomolecular condensate [26,27] that attracts the three proteins through interactions (possibly fortuitous) with intrinsically disordered regions [26–28].

The functional significance of ß-catenin, MafB, and WDR77 association with the region between paused Pol II and the +1 nucleosome is unclear. Depletion of any one of these proteins reduces transformation and affects the transcriptional profile in a similar manner as that observed upon depletion of other transcription factors that affect transformation. However, there is no significant correlation between differential transcription in cells depleted of any of these factors and MafB/WDR77/ß-catenin binding. As mentioned above, the association of these three proteins between paused Pol II and the +1 nucleosome may be fortuitous and hence functionally irrelevant. However, it is also possible that MafB/WDR77/ß-catenin does directly affect the transcription of some genes and/or affects a post-transcriptional process that occurs after Pol II pausing.

## Materials and methods

### Cell lines

MCF-10A-ER-Src cells were grown in DMEM/F12 without phenol red (Thermo Fisher Scientific, 11039–047) + 5% charcoal stripped FBS (Sigma, F6765) containing 1% pen/strep (Thermo Fisher Scientific, 15140122), 20 ng/ml EGF (Peprotech, AF-100-15), 0.5 μg/ml Hydrocortisone (Sigma, H-0888), 0.1 μg/ml cholera toxin (Sigma, C-8052), 10 μg/ml insulin (Sigma, 10516), 2–4 μM AZD0530 (Selleck Chemicals, S1006) as described previously [1,4–6,20]. 1 μM 4 hydroxy tamoxifen (Sigma, H7904) was used to induce the transformation. MDA-MB-231 cells were grown in DMEM (Thermo Fisher Scientific, 11995–073) + 10% FBS (Sigma, TMS-013-B) + 1% pen/strep (Thermo Fisher Scientific, 15140122).

### CRISPR knockout and oligo siRNAs knockdown

We used a CRISPR-blasticidin lentiviral based platform to knockout the genes encoding β-catenin, MafB and WDR77 as described previously [6]. The oligo sequences carried were β-catenin—AAACAGCTCGTTGTACCGCTGGG, WDR77—CCCAAATGCGCCCGCCTGCATGG, and MafB—GCTCAAGTTCGACGTGAAGAAGG. For siRNA knockdowns, the following oligo siRNAs were purchased from Dharmacon, negative control, Cat# D-001810-10-20; β-catenin Cat# L-003482-00; MafB, Cat# L-009018-00; STAT3, Cat# L-003544-00; JUNB, Cat# L-003269-00. These oligo siRNAs were transfected into cells using Lipofectamine RNAiMAX (Thermo Fisher Scientific, 13778050) as described previously [6]. Cells were split 24 hours after oligo siRNA transfection to perform the various assays described below.

### Growth in low attachment conditions assay (GILA)

Cellular transformation was evaluated by growth in low attachment [22]. CRISPR knockout cells were seeded into ultra-low attachment surface 96-well plates (Costar, 3474) for four days, and the amount of ATP was measured by the CellTiter-Glo luminescent cell viability assay (Promega, G7571). As a control for cell proliferation, ATP was measured for the same cells left to grow in regular (high attachment) plates for the same four days.

### Luciferase reporter assay

After siRNA depletion for 24 hours, luciferase reporter plasmids were transfected into cells by TransIT 2020 transfection reagent (Mirus Bio, MIR5400). 24 hours after transfection, cells

were split and treated with either 0.4 μM Tamoxifen (Sigma, H7904) for cell transformation or ethanol as a non-transformed control for 24 h. Firefly and Renilla luciferase activities were determined by Dual-luciferase Reporter Assay kit (Promega, E1910). The STAT3 luciferase reporter plasmid was purchased from Affymetrix, Cat# LR0077. A pGL-AP-1 plasmid containing 6 consensus AP-1 binding sites was used for the evaluation of AP-1 activity and a pRL-CMV plasmid used as an internal control as described previously [6].

## Cell fractionation, co-immunoprecipitation (co-IP), mass spectrometry, and recombinant proteins

Co-IP experiments were performed on separated nuclear and cytoplasmic compartments as described previously [6] using antibodies against β-catenin (BD Bioscience, Cat # 610154), WDR77 (Cell Signaling, Cat# 2018), and MafB (Cell Signaling, Cat# 41019). Western blots were performed as described previously (Ji et al., 2019; He et al. 2021) using antibodies against β-catenin (Cell Signaling, Cat#9581), WDR77 (SCBT, Cat# sc-100899), MafB (SCBT, Cat# sc-10022), MEK1/2 (Bethyl Laboratories, Cat# A302-140A-T), U1 SnRNP70 (SCBT, Cat# sc-9571), and H3 (Abcam, Cat# AB1791). Samples for mass spectrometric analysis were prepared by immunoprecipitating nuclear extracts with an antibody against ß-catenin (Cell Signaling Cat# 8480) and Dynabeads Protein G (ThermoFisher Cat # 10003D). Binding proteins on the Dynabeads were eluted with buffer containing 1.5% SDS, 100 mM Tris pH 7.5, and 10 mM DTT, then precipitated with trichloro acetic acid. Proteins were identified by mass spectrometry performed at the Taplin Mass Spectrometry Facility at Harvard Medical School. For assessing direct interactions, recombinant β-catenin, WDR77, and MafB proteins (expressed from pET plasmids in BL21 *E. coli* cells) were produced as described previously [6] and analyzed by co-IP.

## ChIP-seq and RNA-seq

ChIP-seq were performed as described in previously [4,6]. Briefly, cells were dual cross-linked with 2 mM ethylene glycol bis (succinimidyl succinate) (EGS) and disuccinimidyl glutarate (DSG) and 1% formaldehyde. Chromatin was isolated and then fragmented with 60 units MNase (New England Biolabs, M0247S) at 37 $^{0}$C for 10 minutes. For β-catenin ChIP-seq, cells were first fractionated by Subcellular Protein Fractionation Kit for Cultured Cells (ThermoFisher, Cat# 78840) to obtain a nuclear fraction that was diluted 1:5 with MNase digestion buffer (50 mM HEPES pH 7.9, 140 mM NaCl, 1% Triton X-100, 1 mM CaCl2, 1 mM DTT), and then digested with MNase as above. Antibodies used in the ChIP-seq experiments were against β-catenin (Cell Signaling, Cat#9581), WDR77 (SCBT, Cat# sc-100899), and MafB (SCBT, Cat# sc-10022).

For RNA-seq experiments, mRNeasy Mini Kit (Qiagen, No. 217004) was used for RNA extraction. 0.4 μg total RNA was used for RNA-seq library construction by a TruSeq RNA Library Prep Kit V2 (Illumina, RS-122-2001). Both ChIP-seq and RNA-seq libraries were sequenced at the Bauer Core Facility, Harvard.

## ChIP-seq data analysis

FASTQ reads were aligned to the human reference genome (GRCh38) using Bowtie2 [29]. samtools-1.9 with MAPQ threshold 30 was used to remove low quality reads and picard-tools-2.18 was used to remove PCR duplicates. SPP was used with—cap-num-peak 300000 to call peaks and IDR-2.0.4 [30] with—soft-idr-threshold 0.05 was used to identify peaks conserved across replicates. Correlation coefficients among replicates were for MafB and WDR77 ranged between 0.8 and 0.95. For ß-catenin, where only one replicate passed quality control, pseudo-

replicates were generated by randomly splitting the data in half for further processing. The number of peaks in ethanol (non-transformed) and tamoxifen (transformed) cells were as follows: ß-catenin, 1103 and 3173; MafB, 18091 and 28,503; WDR77, 22,664 and 25,158). Peaks within 2 kb of a GENCODE TSS are considered TSS-proximal; others are considered TSS-distal.

### RNA-seq data analysis

We trimmed adapter sequences, ambiguous 'N' nucleotides (the ratio of "N" > 5%), and low-quality tags (quality score < 20), then aligned trimmed reads against the GENCODE v30 reference transcriptome [31] using STAR [32] with the following parameters:

   —outFilterMultimapNmax 20
   —alignSJoverhangMin 8
   —alignSJDBoverhangMin 1
   —outFilterMismatchNmax 999
   —outFilterMismatchNoverReadLmax 0.04
   —alignIntronMin 20
   —alignIntronMax 1000000
   —alignMatesGapMax 1000000
   —sjdbScore 1

   Counts were normalized to TPM (transcripts per million RNA molecules) using RSEM [33] with the following parameters: "—estimate-rspd—calc-ci." Differential expression analysis was performed with DESeq2 [34] with default parameters. Separate DESeq2 runs were performed for siRNA against each factor, comparing replicates treated with the siRNA against the given factor versus replicates treated with the control siRNA. DESeq2 was performed independently for tamoxifen- and ethanol-treated cells for each siRNA. Genes with a multiple-testing adjusted $p$-value <0.01 were considered to be differentially expressed.

### Histone and Pol-II signal aggregation

We downloaded Pol-II ChIP-seq signal in tamoxifen-treated MCF-10A cells from the ENCODE Portal (accession ENCFF114YIB). We then identified all protein-coding GENCODE v30 TSSs within 2 kb of a ß-catenin peak using bedtools intersect. We aligned these TSSs, oriented them according to strand, and computed the average ChIP-seq signals for ß-catenin, MafB, WDR77, H3K4me3, H3K27ac, and Pol-II from our datasets and the ENCODE dataset around these TSSs using the pyBigWig function in the deepTools2 package [35].

### Data deposition

All sequencing data were deposited on National Cancer for Biotechnology Information Gene Expression Omnibus (GEO). GSE195740 is the accession number for all the data, with GSE1957437 being the subset for the ChIP-seq data and GSE195739 for the RNA-seq data.

## Supporting information

**S1 Raw images.**
(PDF)

## Author Contributions

**Conceptualization:** Lizhi He, Kevin Struhl.

**Data curation:** Lizhi He, Mingshi Gao, Henry Pratt.

**Formal analysis:** Lizhi He, Mingshi Gao, Henry Pratt, Zhiping Weng, Kevin Struhl.

**Funding acquisition:** Zhiping Weng, Kevin Struhl.

**Investigation:** Lizhi He, Mingshi Gao, Henry Pratt.

**Project administration:** Zhiping Weng, Kevin Struhl.

**Software:** Mingshi Gao, Henry Pratt.

**Supervision:** Zhiping Weng, Kevin Struhl.

**Validation:** Lizhi He.

**Visualization:** Mingshi Gao, Henry Pratt.

**Writing – original draft:** Lizhi He, Mingshi Gao, Henry Pratt, Zhiping Weng, Kevin Struhl.

**Writing – review & editing:** Lizhi He, Mingshi Gao, Henry Pratt, Zhiping Weng, Kevin Struhl.

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
