## [Decision Letter · Decision Letter 0]

18 Mar 2022

PONE-D-22-04674MafB, WDR77, and ß-catenin interact with each other and have similar genome association profilesPLOS ONE

Dear Dr. Struhl,

thank you for submitting your manuscript to PLOS ONE. After careful consideration, and having read the manuscript myself, I concur with the Reviewer that it has many merits but does not fully meet PLOS ONE’s publication criteria as it currently stands. Specifically, I would focus on point 1 raised by the Reviewer. Point 2 would be a plus, but I don't think it is mandatory, as the data currently stands.

Thereafter, I invite you to resubmit a revised version of the manuscript.

Please submit your revised manuscript by May 02 2022 11:59PM. Please include the following items when submitting your revised manuscript:A rebuttal letter that responds to each point raised by the academic editor and reviewer(s). You should upload this letter as a separate file labeled 'Response to Reviewers'.A marked-up copy of your manuscript that highlights changes made to the original version. You should upload this as a separate file labeled 'Revised Manuscript with Track Changes'.An unmarked version of your revised paper without tracked changes. You should upload this as a separate file labeled 'Manuscript'.If applicable, we recommend that you deposit your laboratory protocols in protocols.io to enhance the reproducibility of your results. Protocols.io assigns your protocol its own identifier (DOI) so that it can be cited independently in the future. For instructions see: https://journals.plos.org/plosone/s/submission-guidelines#loc-laboratory-protocols. Additionally, PLOS ONE offers an option for publishing peer-reviewed Lab Protocol articles, which describe protocols hosted on protocols.io. Read more information on sharing protocols at https://plos.org/protocols?utm_medium=editorial-email&utm_source=authorletters&utm_campaign=protocols.

We look forward to receiving your revised manuscript.

Kind regards,

Roberto Mantovani

Academic Editor

PLOS ONE

Journal Requirements:

[This work was supported by grants from the National Institutes of Health to KS (CA 107486) and ZW (HG009446).]

 [HHS | National Institutes of Health (NIH):Lizhi He,Kevin Struhl CA107486; HHS | National Institutes of Health (NIH):Mingshi Gao,Henry Pratt,Zhiping Weng HG009446]

Reviewers' comments:

Reviewer's Responses to Questions

**Comments to the Author**

1. Is the manuscript technically sound, and do the data support the conclusions?

Reviewer #1: Yes

2. Has the statistical analysis been performed appropriately and rigorously? 

Reviewer #1: No

3. Have the authors made all data underlying the findings in their manuscript fully available?

Reviewer #1: Yes

4. Is the manuscript presented in an intelligible fashion and written in standard English?

Reviewer #1: Yes

5. Review Comments to the Author

Reviewer #1: The manuscript by He et al. describes the physical and genomic association between MafB, WDR77 and β-catenin, three proteins belonging to different families that have been previously reported to be involved in cellular transformation processes. This is a descriptive work by an excellent team with a solid expertise in genomic studies. It is not clear to me whether this a resubmission to Plos One journal or a transfer from another journal, as a Rebuttal letter has been attached by the Authors. Anyway, I have read the answers to Referees’ comments and I agree with the Authors that these results could be useful for the research activity of other scientists studying the role of MafB, WDR77 and β-catenin proteins. In the absence of functional data that would allow to submit the manuscript to higher impact journals, Plos One is the appropriate journal.

1. I respect the frankness of the Authors that stated that the rationale of the work rose from unrelated observations on β-catenin interactors and MafB. Despite this, I believe that the first paragraph of the Results section does not give the reader a clear idea about the research. The Authors reported the reference to their previous works on YAP/TAZ and S100A8/A9, but, as they stated in their Rebuttal, these works are not linked to the present one. I believe that the first paragraph should be differently elaborated to give the reader a better understanding of the work.

In the first and third paragraphs, the Authors refer to mass spectrometry experiment that allows to identify WDR77 as β-catenin interactor. Are these results published or can the Author provide some details?

2. I believe that the results would be of great interest to other scientists if the Authors could identify transcription factor binding sites within MafB, WDR77 and β-catenin ChIP-Seq data. Moreover, it would be useful if the Authors could add GO/KEGG analysis of RNA seq profiles. In particular, which are gene categories of differentially expressed genes overlapping between the three transcriptional profiles from RNAi cells showed in Figure 5A-B?

Minor:

-The MCF10A-ER-SRC cellular model has been previously used and described by the Authors, but I suggest to add at least that tamoxifen treatment is used to induce Src-mediated transformation (second paragraph of the results section:” …in the presence of tamoxifen”) to help readers that did not read previous works.

- Statistical analysis should be added for the histograms showed in Fig. 1D-E.

- Page 7: the Authors described H3K4me3 and H3K27ac as markers of +1 nucleosome position but they show only H3K4me3 Fig.4C; I suggest to remove H3K27ac from the description.

- Please, edit “CRIPSR” in Figure 1A, and “ß-catenin siRN” in Fig. 1D, “noramlized ChIP-seq signal” Fig. 3A.

6. PLOS authors have the option to publish the peer review history of their article (what does this mean?). If published, this will include your full peer review and any attached files.

Reviewer #1: No

---

## [Author Response · Author response to Decision Letter 0]

6 Apr 2022

Dear Editors,

Attached is a revised version of our paper still entitled “MafB, WDR77, and ß-catenin interact with each other and have similar genome association profiles” (RC2021-01184) for publication in PLOS ONE. I should clarify that the original manuscript was sent to Review Commons and was then transferred to PLOS ONE. The original response was to the reviewers of the manuscript submitted to Review Commons. Here, the response is to the single reviewer of the transferred PLOS ONE manuscript, who suggested a few very minor changes, which we have heeded. Point-by-point responses below.

Reviewer 1

1. We agree that readers should not have to go to our previous papers to understand technical aspects of our model. As requested, we added several sentences at the beginning of the results section (first paragraph) that described the key features of the model. The mass spectrometry experiments are now described in the methods, and a table of interacting proteins is now presented (this was an unintended omission for the previous version).

2. Most of the binding sites are those described in detail in the paper. We don’t think it is particularly useful to discuss the other sites as we haven’t learned much. As for the 

GO/KEGG analysis of regulated genes, we pointed out in our previous response that the expression profiles in the ß-catenin, WDR77, MafB, and TCF4 depletion experiments are similar to each other and to expression profiles in depletions of other proteins important for transformation in our model. This is because anything that affects the positive feedback loop involved in transformation gives very similar gene expression profiles. This has been extensively discussed in our previous papers (cited in the present paper), and indeed GO/KEGG analysis has been done before. In terms of transcriptional and functional pathways, the results in this paper do not add anything beyond the fact that ß-catenin, WDR77, MafB, and TCF4 are part of the transformation pathway.

Minor points: All as requested. We added details about tamoxifen in both the results and methods section. I don’t know what statistical analysis is desired in Fig. 1D,E; we show error bars and ±SD of 3 replicates. We deleted H3K27ac from the description of Fig. 4C. We fixed the typographical errors in the Figures (I’m impressed the Reviewer noticed them; we didn’t).

---

## [Editor Report · Decision Letter 1]

18 Apr 2022

MafB, WDR77, and ß-catenin interact with each other and have similar genome association profiles

PONE-D-22-04674R1

Dear Dr. Prof. Struhl,

We’re pleased to inform you that your manuscript has been judged scientifically suitable for publication and will be formally accepted for publication once it meets all outstanding technical requirements.

Kind regards,

Roberto Mantovani

Academic Editor

PLOS ONE
---

## [Editor Report · Acceptance letter]

20 Apr 2022

PONE-D-22-04674R1 

MafB, WDR77, and ß-catenin interact with each other and have similar genome association profiles 

Dear Dr. Struhl:

I'm pleased to inform you that your manuscript has been deemed suitable for publication in PLOS ONE. Congratulations! Your manuscript is now with our production department. 

Kind regards, 

on behalf of

Prof. Roberto Mantovani 

Academic Editor

PLOS ONE